# The $\Delta$P Programming Language: Combining Imperative, Logical, and Probabilistic Capabilities

**Sergey Goncharov**[1*], **Andrey Nechesov**[2,3,4], **Igor Anureev**[5], **Dmitry Kondratyev**[5], **Evgenii Vityaev**[2,3], Natalia Garanina[5], Ivan Gorobets[2], Dmitry Sviridenko[2], Gulnara Yakhyaeva[2]

[1]*Sobolev Institute of Mathematics, 630090 Novosibirsk, Russia*
[2]*Artificial Intelligence Center, Novosibirsk State University, 630090 Novosibirsk, Russia*
[3]*International Artificial Intelligence Committee, (IAIC), Dubai, UAE*
[4]*Russian Engineering Academy (IAE), Moscow, Russia*
[5]*A.P. Ershov Institute of Informatics Systems, Novosibirsk, Russia*
*Corresponding author: s.novikov1@g.nsu.ru*

## Abstract

Probabilistic reasoning is an essential component of modern intelligent systems, especially in domains where uncertainty, incomplete knowledge, and stochastic processes must be modeled explicitly. This paper presents the $\Delta$P programming language, a probabilistic extension of an imperative programming framework designed to support reasoning with both deterministic and probabilistic predicates. The language combines traditional programming constructs with mechanisms for expressing probabilistic knowledge and uncertainty in program logic. We describe the syntax and core constructs of $\Delta$P, including constants, terms, predicates, declarations, and control statements. In addition, we formalize the operational semantics of the language, providing a precise definition of program execution and probabilistic evaluation. The paper also introduces the internal logical framework used for probabilistic reasoning within programs. A case study and several illustrative examples demonstrate how the proposed language can be used to model and analyze problems involving uncertainty. The presented approach aims to bridge the gap between classical imperative programming and probabilistic reasoning, offering a structured framework for building intelligent systems that require integrated logical and probabilistic computation.

## 1 Introduction

Many modern computational systems must operate in environments characterized by uncertainty, incomplete information, and stochastic behavior. Traditional programming languages provide powerful mechanisms for deterministic computation, but they often lack native constructs for expressing probabilistic knowledge and reasoning about uncertain events. As a result, developers frequently rely on external libraries or ad-hoc mechanisms to incorporate probabilistic reasoning into programs, which can complicate both program design and formal analysis.

Probabilistic programming languages aim to address this limitation by integrating probabilistic concepts directly into the programming model. Such languages allow developers to describe uncertain processes, probabilistic dependencies, and stochastic decision making in a unified framework. At the same time, formal semantics are necessary to ensure that probabilistic constructs behave predictably and can be analyzed rigorously.

This paper introduces the $\Delta$P programming language, an imperative language designed to support both standard logical predicates and probabilistic predicates within a unified programming environment. The language provides constructs for defining constants, terms, formulas, declarations, and control structures while allowing probabilistic information to be represented and processed directly within programs.

We present the core structure of the language and define its operational semantics, which formally describe how programs are executed and how probabilistic predicates are evaluated. In addition, we discuss the internal logic used to support probabilistic reasoning in $\Delta$P programs. To illustrate the practical applicability of the language, we provide examples and a case study demonstrating how $\Delta$P can be used to model problems involving uncertainty and probabilistic inference.

The main contribution of this work is the design and formalization of a programming framework that integrates imperative programming constructs with probabilistic reasoning capabilities. This integration enables clearer program structure, more expressive modeling of uncertain processes, and a foundation for further research in probabilistic programming and reasoning systems.

## 2 THE $\Delta$P PROGRAMMING LANGUAGE

The language $\Delta$P is an untyped imperative language with standard and probabilistic predicates.

We define the main language constructs: constants, terms, formulas, declarations, statements, macros, and programs.

### 2.1 CONSTANTS

Let $c$ denote constants.

Constants include:

- integers $-3$, $0$, $7$, etc.;
- real numbers $-3.42$, $0.0$, $27.23$, etc.;
- strings "abc", etc.;
- boolean values `true` and `false`;
- lists $[c_1, \ldots, c_n]$;
- range constants $[c_1 : c_2]$.

### 2.2 BUILT-IN PREDICATES

Since $\Delta$P is an untyped language, the following built-in predicates are used to model types:

- `nat`$(x)$: $x$ is a natural number, including `0`;
- `int`$(x)$: $x$ is an integer;
- `float`$(x)$: $x$ is a real number;
- `bool`$(x)$: $x$ is a boolean value;
- `string`$(x)$: $x$ is a string;
- `list`$(x)$: $x$ is a list;
- `range`$(x)$: $x$ is an integer range.

### 2.3 TERMS

Let $t$ denote terms.

Terms are divided into:

- term constants;
- variables $x$;
- unary terms $\circ t$, where $\circ$ may be the arithmetic operation $-$;
- binary terms $t_1 \circ t_2$, where $\circ$ may be one of the arithmetic operations $+$, $-$, $*$, $/$, $\%$;
- parenthesized terms $(t)$ for grouping expressions;
- list expressions $[t_1, \ldots, t_n]$, representing lists;
- range expressions $[t_1 : t_2]$, representing lists of integers from $t_1$ to $t_2$.

## 2.4 FORMULAS

Let $r$ denote formulas, and $q$ denote predicate names.

Formulas are divided into:

- float constants from 0 to 1 (probabilities of the true value);
- unary formulas $\circ r$, where $\circ$ may be the logical operator `~`;
- binary formulas $r_1 \circ r_2$, where $\circ$ may be one of the logical operators `&`, `|`, `->`;
- comparison formulas $t_1 \circ t_2$, where $\circ$ may be one of the comparison operators `=`, `!=`, `<`, `<=`, `>`, `>=`;
- membership formulas $t_1$ `in` $t_2$, where $t_2$ is a list or range;
- parenthesized formulas $(r)$ for grouping;
- quantified formulas `!` $x$ `:` $t$ `(`$r$`)` and `?` $x$ `:` $t$ `(`$r$`)`, where $x$ is the quantified variable, $t$ is a list or range restricting the values of this variable, and $r$ is the subformula;
- predicate formulas $q$`(`$t_1$`,` `...,` $t_n$`)`, returning the result of applying the predicate $q$ to the arguments $t_1, ..., t_n$.

## 2.5 DECLARATIONS

Let $d$ denote declarations.

Declarations are divided into:

- static predicate definitions `sp` $q(x_1, ..., x_n)$ `:=` $r$`;`, where the formula $r$, depending on variables $x_1, ..., x_n$, specifies the definition of predicate $q$;
- probabilistic predicate definitions `dp` $q(x_1, ..., x_n)$ `:` $r$`;`, where the formula $r$, depending on variables $x_1, ..., x_n$, specifies the domain of predicate $q$. This formula does not contain dynamic predicates. The values of dynamic predicates are defined using a database that stores the values of these predicates for specific arguments.

## 2.6 STATEMENTS

Let $s$ denote statements.

Statements are divided into:

- declarations $d$;
- assignment statements $x$ `:=` $t$;
- `input(`$x_1$`,` `...,` $x_n$`);` – inputs the values, storing them in variables $x_1, ..., x_n$;
- `output(`$e_1$`,` `...,` $e_n$`);` – outputs the values of expressions $e_1, ..., e_n$;
- conditional statements `if` `(`$r$`)` `then` $s_1$ `else` $s_2$;
- for statements `for` $x$ `in` $e$ `do` $s$;
- block statements $\{s_1 ... s_n\}$, grouping sequences of statements;
- probabilistic predicate update statements $q$`(`$t_1$`,` `...,` $t_n$`)` `:=` `true;`, $q$`(`$t_1$`,` `...,` $t_n$`)` `:=` `false;`, and $q$`(`$t_1$`,` `...,` $t_n$`)` `:=` `undef;`.

## 2.7 MACROS

Let $u$ denote macros.

Macros include:

- `#and` $x$ $y$ `:=` $t$; – defines the probability of conjunction in terms of the probabilities $x$ and $y$ of its operands (default value $x * y$);

- #or $x$ $y$ := $t$; – defines the probability of disjunction in terms of the probabilities $x$ and $y$ of its operands (default value $x + y - x * y$);

- #not $x$ := $t$; – defines the probability of negation in terms of the probability $x$ of its operand (default value $1 - x$);

- #imply $x$ $y$ := $t$; – defines the probability of implication in terms of the probabilities $x$ and $y$ of its operands (default value $1 - x + x * y$);

- #pmode := $w$;, where $w$ takes one of the values `decision` or `simulation`, – specifies the probability evaluation mode (default value `decision`).

## 2.8 PROGRAM

A program has the form: $u_1 \dots u_m \ s_1 \dots s_n$, i.e., a sequence of macros followed by a sequence of statements.

## 3 OPERATIONAL SEMANTICS OF THE $\Delta$P LANGUAGE

We define a formal small-step operational semantics for the language $\Delta$P, given as a labeled transition system $(\Sigma, L, \rightarrow_l)$, where $\Sigma$ is the set of program states, $L$ is the set of $\Delta$P instructions (labels), and $\rightarrow_l$ is a family of binary relations on $\Sigma$ (transition relations).

Let $N$ be the set of names that variables and predicates may take (for a given program, the sets of variable names, static predicate names, and dynamic predicate names are pairwise disjoint), $V$ the set of values that variables may take, and $R$ the set of formulas.

A state $\sigma$ consists of the following components:

- $\sigma.v \subseteq N$ — the set of variables defined in the program in state $\sigma$;

- $\sigma.v.val \in \sigma.v \rightarrow V$ — the valuation function for variables, returning their values in $\sigma$;

- $\sigma.sp \subseteq N$ — the set of names of static predicates in $\sigma$;

- $\sigma.sp.arg \in \sigma.sp \rightarrow R$ — a function returning the list of arguments of static predicates in $\sigma$;

- $\sigma.sp.def \in \sigma.sp \rightarrow R$ — the valuation function for static predicates, returning their definitions in $\sigma$;

- $\sigma.dp \subseteq N$ — the set of names of dynamic predicates in $\sigma$;

- $\sigma.dp.arg \in \sigma.dp \rightarrow R$ — a function returning the list of arguments of dynamic predicates in $\sigma$;

- $\sigma.dp.dom \in \sigma.dp \rightarrow R$ — the valuation function for dynamic predicates, returning their domain definitions in $\sigma$;

- $\sigma.db \in N \rightarrow V^* \rightarrow \{T, F, U\}$ — a database storing the values of dynamic predicates for specific arguments in $\sigma$. The values $T$, $F$, and $U$ correspond to the cases where the predicate is true, false, and undefined, respectively;

- $\sigma.mode \in \{decision, simulation\}$ — determines the program execution mode in $\sigma$ according to the #pmode macro in $\sigma$;

- $\sigma.not \in V \rightarrow V$ — a function implementing negation according to the #not macro in $\sigma$;

- $\sigma.and \in V \times V \rightarrow V$ — a function implementing conjunction according to the #and macro in $\sigma$;

- $\sigma.or \in V \times V \rightarrow V$ — a function implementing disjunction according to the #or macro in $\sigma$;

- $\sigma.imply \in V \times V \rightarrow V$ — a function implementing implication according to the #imply macro in $\sigma$;

- $\sigma.in$ — stores the list of values that have been get from input in $\sigma$;

- $\sigma.out$ — stores the list of values that have been sent to output in $\sigma$;

- $\sigma.val$ — stores the last computed value in $\sigma$.

Let $\sigma[y_1 := v_1, \ldots, y_n := v_n]$ denote the state obtained by replacing the values of components $y_1, \ldots, y_n$ with $v_1, \ldots, v_n$, respectively.

Let $f[y_1 := v_1, \ldots, y_n := v_n]$ denote the function obtained by replacing the values of $f$ at arguments $y_1, \ldots, y_n$ with $v_1, \ldots, v_n$, respectively.

Let $r[x_1/t_1, \ldots, x_n/t_n]$ denote the result of substituting variables $x_1, \ldots, x_n$ with terms $t_1, \ldots, t_n$ in the formula $r$.

Let $\sigma.dp.arg = [x_1, \ldots, x_n]$.

The function $prob(q, \sigma)$, which computes the probability of a dynamic predicate $q$ in state $\sigma$, is defined as

$$\frac{|Pos|}{|Pos| + |Neg|}$$

where:

- $Pos$ is the set of $[v_1, \ldots, v_n]$ such that $\sigma \to_{\sigma.dp.dom(q)[x_1/v_1, \ldots, x_n/v_n]} \sigma'$, $\sigma'.val = 1$, and $\sigma.db(q)([v_1, \ldots, v_n]) = T$;
- $Neg$ is the set of $[v_1, \ldots, v_n]$ such that $\sigma \to_{\sigma.dp.dom(q)[x_1/v_1, \ldots, x_n/v_n]} \sigma'$, $\sigma'.val = 1$, and $\sigma.db(q)([v_1, \ldots, v_n]) = F$.

In the case where $|Pos| + |Neg| = 0$, we define $prob(q, \sigma) = 0.5$.

Let the function $rand(v)$ for $0 \le v \le 1$ model a random coin toss, where the probability of heads is $v$ and tails is $1 - v$. The function returns 1 if heads occurs and 0 if tails occurs.

We define the left fold function $foldl$ over a binary operation in the usual way:

- $foldl(\circ, [v_1]) = v_1$;
- $foldl(\circ, [v_1, v_2, \ldots, v_n]) = foldl(\circ, [v_1 \circ v_2, \ldots, v_n])$.

The subsections below define the transition relation for the various kinds of $\Delta$P instructions.

## 3.1 CONSTANTS

The transition relation for constants is defined by the following axiom:

$$\sigma \to_c \sigma[val := c].$$

## 3.2 BUILT-IN PREDICATES

The transition relation for built-in predicates is given by the following rules:

$$\frac{\sigma \to_{t_1} \sigma_1 \quad \ldots \quad \sigma \to_{t_n} \sigma_n \quad q(\sigma_1.val, \ldots, \sigma_n.val)}{\sigma \to_{q(t_1, \ldots, t_n)} \sigma[val := 1]};$$

$$\frac{\sigma \to_{t_1} \sigma_1 \quad \ldots \quad \sigma \to_{t_n} \sigma_n \quad \neg q(\sigma_1.val, \ldots, \sigma_n.val)}{\sigma \to_{q(t_1, \ldots, t_n)} \sigma[val := 0]}.$$

## 3.3 TERMS

The transition relation for terms is defined by the following rules:

$$\frac{x \in \sigma.v}{\sigma \to_x \sigma[val := \sigma.v.val(x)]};$$

$$\frac{\sigma \to_t \sigma'}{\sigma \to_{\circ t} \sigma[val := \circ(\sigma'.val)]};$$

$$\frac{\sigma \to_{t_1} \sigma_1 \quad \sigma \to_{t_2} \sigma_2 \quad (\sigma_1.val) \circ (\sigma_2.val) \text{ is defined}}{\sigma \to_{t_1 \circ t_2} \sigma[val := (\sigma_1.val) \circ (\sigma_2.val)]};$$

$$\frac{\sigma \to_t \sigma'}{\sigma \to_{(t)} \sigma'};$$

$$\frac{\sigma \to_{t_1} \sigma_1 \quad \ldots \quad \sigma \to_{t_n} \sigma_n}{\sigma \to_{[t_1, \ldots, t_n]} \sigma[val := [\sigma_1.val, \ldots, \sigma_n.val]]};$$

$$\frac{\sigma \to_{t_1} \sigma_1 \quad \sigma \to_{t_2} \sigma_2 \quad \sigma_1.val \le \sigma_2.val}{\sigma \to_{t_1:t_2} \sigma[val := [\sigma_1.val, \ldots, \sigma_2.val]]}.$$

### 3.4 FORMULAS

Let $sem(\circ, s)$ return $s.not$, $s.and$, $s.or$, or $s.imply$ depending on the operation $\circ$ (i.e., its semantic interpretation).

The transition relation for formulas is defined by the following rules:

$$\frac{\sigma \to_r \sigma'}{\sigma \to_{\sim r} \sigma[val := sem(\sim, \sigma')(\sigma'.val)]};$$

$$\frac{\sigma \to_{r_1} \sigma_1 \quad \sigma \to_{r_2} \sigma_2}{\sigma \to_{r_1 \circ r_2} \sigma[val := sem(\circ, \sigma)(\sigma_1.val, \sigma_2.val)]};$$

$$\frac{\sigma \to_{t_1} \sigma_1 \quad \sigma \to_{t_2} \sigma_2}{\sigma \to_{t_1 \circ t_2} \sigma[val := (\sigma_1.val) \circ (\sigma_2.val)]};$$

$$\frac{\sigma \to_r \sigma'}{\sigma \to_{(r)} \sigma'};$$

$$\frac{\sigma \to_t \sigma' \quad \sigma'.val = [c_1, \ldots, c_n] \quad \sigma \to_{r[x/c_1]} \sigma_1 \quad \ldots \quad \sigma \to_{r[x/c_n]} \sigma_n}{\sigma \to_{!x:t(r)} \sigma[val := foldl(\sigma.and, [\sigma_1.val, \ldots, \sigma_n.val])]};$$

$$\frac{\sigma \to_t \sigma' \quad \sigma'.val = [c_1, \ldots, c_n] \quad \sigma \to_{r[x/c_1]} \sigma_1 \quad \ldots \quad \sigma \to_{r[x/c_n]} \sigma_n}{\sigma \to_{?x:t(r)} \sigma[val := foldl(\sigma.or, [\sigma_1.val, \ldots, \sigma_n.val])]};$$

$$\frac{\begin{array}{c} p \in \sigma.sp \quad \sigma.sp.arg(q) = [x_1, \ldots, x_n] \\ \sigma \to_{t_1} \sigma_1 \quad \ldots \quad \sigma \to_{t_n} \sigma_n \\ \sigma \to_{(\sigma.sp.def(q))[x_1/\sigma_1.val, \ldots, x_n/\sigma_n.val]} \sigma' \end{array}}{\sigma \to_{q(t_1, \ldots, t_n)} \sigma'};$$

$$\frac{\begin{array}{c} p \in \sigma.dp \quad \sigma.dp.arg(q) = [x_1, \ldots, x_n] \\ \sigma.mode = decision \\ \sigma \to_{t_1} \sigma_1 \quad \ldots \quad \sigma \to_{t_n} \sigma_n \\ \sigma \to_{(\sigma.dp.dom(q))[x_1/\sigma_1.val, \ldots, x_n/\sigma_n.val]} \sigma' \quad \sigma'.val = 1 \end{array}}{\sigma \to_{q(t_1, \ldots, t_n)} \sigma[val := prob(p, \sigma)]};$$

$$\frac{\begin{array}{c} p \in \sigma.dp \quad \sigma.dp.arg(q) = [x_1, \ldots, x_n] \\ \sigma.mode = decision \\ \sigma \to_{t_1} \sigma_1 \quad \ldots \quad \sigma \to_{t_n} \sigma_n \\ \sigma \to_{(\sigma.dp.dom(q))[x_1/\sigma_1.val, \ldots, x_n/\sigma_n.val]} \sigma' \quad \sigma'.val = 0 \end{array}}{\sigma \to_{q(t_1, \ldots, t_n)} \sigma[val := 0.5]};$$

$$p \in \sigma.dp \quad \sigma.dp.arg(q) = [x_1, \ldots, x_n]$$
$$\sigma.mode = simulation$$
$$\sigma \to_{t_1} \sigma_1 \quad \ldots \quad \sigma \to_{t_n} \sigma_n$$
$$\frac{\sigma \to_{(\sigma.dp.dom(q))[x_1/\sigma_1.val,\ldots,x_n/\sigma_n.val]} \sigma' \quad \sigma'.val = 1}{\sigma \to_{q(t_1,\ldots,t_n)} \sigma[val := rand(prob(p,\sigma))]};$$

$$p \in \sigma.dp \quad \sigma.dp.arg(q) = [x_1, \ldots, x_n]$$
$$\sigma.mode = simulation$$
$$\sigma \to_{t_1} \sigma_1 \quad \ldots \quad \sigma \to_{t_n} \sigma_n$$
$$\frac{\sigma \to_{(\sigma.dp.dom(q))[x_1/\sigma_1.val,\ldots,x_n/\sigma_n.val]} \sigma' \quad \sigma'.val = 0}{\sigma \to_{q(t_1,\ldots,t_n)} \sigma[val := rand(0.5)]}.$$

## 3.5 DECLARATIONS

The transition relation for declarations is defined by the following rules:

$$\frac{q \notin \sigma.v \cup \sigma.dp}{\sigma \to_{sp \ q(x_1,\ldots,x_n) := r;}};$$
$$\sigma[sp := \sigma.sp \cup \{q\},$$
$$sp.arg := \sigma.sp.arg[q := [x_1,\ldots,x_n]],$$
$$sp.def := \sigma.sp.def[q := r]]$$

$$\frac{q \notin \sigma.v \cup \sigma.sp}{\sigma \to_{dp \ q(x_1,\ldots,x_n) := r;}}.$$
$$\sigma[dp := \sigma.dp \cup \{q\},$$
$$dp.arg := \sigma.dp.arg[q := [x_1,\ldots,x_n]],$$
$$dp.dom := \sigma.dp.dom[q := r]]$$

## 3.6 STATEMENTS

The transition relation for statements is defined by the following rules:

$$\frac{x \notin \sigma.sp \cup \sigma.dp \quad \sigma \to_t \sigma'}{\sigma \to_{x \ := \ t;} \sigma'[v := \sigma.v \cup \{x\}, v.val := (\sigma'.v.val)[x := \sigma'.val]]};$$

$$\frac{\{x_1,\ldots,x_n\} \nsubseteq \sigma.sp \cup \sigma.dp \quad c_1,\ldots,c_n \in C \quad \sigma \to_{\{x_1:=c_1; \ldots \ x_n:=c_n;\}} \sigma'}{\sigma \to_{input(x_1,\ldots,x_n);} \sigma'[in := \sigma.in + [c_1,\ldots,c_n]]};$$

$$\frac{\sigma \to_{t_1} \sigma_1 \quad \ldots \quad \sigma \to_{t_n} \sigma_n}{\sigma \to_{output(t_1,\ldots,t_n);} \sigma[out := \sigma.out + [\sigma_1.val,\ldots,\sigma_n.val]]};$$

$$\frac{\sigma \to_r \sigma' \quad \sigma'.val \geq 0.5 \quad \sigma \to_{s_1} \sigma''}{\sigma \to_{if \ (r) \ then \ s_1 \ else \ s_2} \sigma''};$$

$$\frac{\sigma \to_r \sigma' \quad \sigma'.val < 0.5 \quad \sigma \to_{s_2} \sigma''}{\sigma \to_{if \ (r) \ then \ s_1 \ else \ s_2} \sigma''};$$

$$\frac{\sigma \to_e \sigma' \quad \sigma' \to_{for1 \ x \ in \ \sigma'.val \ do \ s} \sigma''}{\sigma \to_{for \ x \ in \ e \ do \ s} \sigma''};$$

$$\frac{c = []}{\sigma \to_{for1\ x\ in\ c\ do\ s} \sigma};$$

$$\frac{c = [c_1 : c_2] \quad c_1 > c_2}{\sigma \to_{for1\ x\ in\ c\ do\ s} \sigma};$$

$$\frac{c = [c_1, c_2, ..., c_n] \quad \sigma \to_{\{x:=c_1;\ s\}} \sigma' \quad \sigma' \to_{for1\ x\ in\ [c_2,...,c_n]\ do\ s} \sigma''}{\sigma \to_{for1\ x\ in\ c\ do\ s} \sigma''};$$

$$\frac{c = [c_1 : c_2] \quad c_1 \le c_2 \quad \sigma \to_{\{x:=c_1;\ s\}} \sigma' \quad \sigma' \to_{for1\ x\ in\ [c_1+1:c_2]\ do\ s} \sigma''}{\sigma \to_{for1\ x\ in\ c\ do\ s} \sigma''};$$

$$\frac{\sigma \to_{s_1} \sigma_1 \quad \cdots \quad \sigma_{n-1} \to_{s_n} \sigma_n}{\sigma \to_{\{s_1...s_n\}} \sigma_n};$$

$$\frac{\sigma \to_{t_1} \sigma_1 \quad \cdots \quad \sigma \to_{t_n} \sigma_n}{\sigma \to_{q(t_1,...,t_n):=true;} \sigma[db := \sigma.db[q := \sigma.db(q)[[\sigma_1.val, \ldots, \sigma_n.val] := T]]]};$$

$$\frac{\sigma \to_{t_1} \sigma_1 \quad \cdots \quad \sigma \to_{t_n} \sigma_n}{\sigma \to_{q(t_1,...,t_n):=false;} \sigma[db := \sigma.db[q := \sigma.db(q)[[\sigma_1.val, \ldots, \sigma_n.val] := F]]]};$$

$$\frac{\sigma \to_{t_1} \sigma_1 \quad \cdots \quad \sigma \to_{t_n} \sigma_n}{\sigma \to_{q(t_1,...,t_n):=undef;} \sigma[db := \sigma.db[q := \sigma.db(q)[[\sigma_1.val, \ldots, \sigma_n.val] := U]]]}.$$

## 3.7 MACROS

The transition relation for macros is defined by the following axioms:

$$\sigma \to_{\#not\ x\ :=\ t;} \sigma[not := (\lambda x.\ t)];$$

$$\sigma \to_{\#and\ x\ y\ :=\ t;} \sigma[and := (\lambda x\ y.\ t)];$$

$$\sigma \to_{\#or\ x\ y\ :=\ t;} \sigma[or := (\lambda x\ y.\ t)];$$

$$\sigma \to_{\#imply\ x\ y\ :=\ t;} \sigma[imply := (\lambda x\ y.\ t)];$$

$$\sigma \to_{\#pmode\ x;} \sigma[mode := x].$$

Here $(\lambda x_1 \ldots x_n.\ t)$ denotes a $\lambda$-function with arguments $x_1, \ldots, x_n$.

## 3.8 PROGRAM

The transition relation for a program is defined by the following rule:

$$\frac{\begin{array}{ccc} \sigma \to_{u_1} \sigma_1 & \cdots & \sigma_{m-1} \to_{u_m} \sigma_m \\ \sigma_m \to_{s_1} \sigma_{m+1} & \cdots & \sigma_{m+n-1} \to_{s_n} \sigma_{m+n} \end{array}}{\sigma \to_{u_1\ ...\ u_m\ s_1\ ...\ s_n} \sigma_{m+n}}.$$

## 4    CASE STUDY

To illustrate the practical utility of the $\Delta$P language, we present two realistic scenarios from the domain of smart city management. The first example demonstrates the use of the simulation mode for constructing a digital twin of urban traffic. The second example employs the decision mode to support resource allocation for emergency services. Both examples highlight key features of $\Delta$P: probabilistic predicates, domain definitions, and flexible control over the evaluation mode.

### 4.1    EXAMPLE 1: TRAFFIC FLOW PREDICTION (SIMULATION MODE)

Modern smart cities deploy networks of sensors that monitor traffic intensity at key intersections. Using data from these sensors, it is possible to provide drivers with information about potential traffic congestion via roadside information boards.

The program below defines a small fragment of a digital twin of the urban traffic system. It uses the dynamic predicate `trafficJam`, which represents traffic congestion statistics in winter depending on the area, month, time, and weather. The simulation mode (`#pmode simulation`) allows us to draw random samples from the posterior distribution given sensor readings. Macros such as `#and`, `#or`, etc., remain at their default values.

```
#pmode simulation;

Area := [1:15];
rainy := 1; normal := 2; snowy := 3;
Weather := [rainy:snowy];
Time := [0:23];
Month := [1:12];
inHours := 2;

sp winter(month) := month = 12 | month = 1 | month = 2;

input(curA, curM, curT, curW, inHours);

if(curA in Area & winter(curM) & curT in Time &
    curT + inHours <= 23 & curW in Weather & inHours in [2:5])
then {
  dp trafficJam(a, m, t, w) : a = curA & winter(m) &
      curT + inHours - 1 <= t <= curT + inHours & w = curW;

  if(trafficJam(curA, curM, curT, curW))
  then output("Traffic congestion may occur in Area ", curA,
    " in ", inHours, " hours.");
  else output("Traffic congestion is not expected in Area ", curA,
    " in ", inHours, " hours.";
}
else output("The data is incorrect.")
```

Four domains are defined in the program: `Area`, `Weather`, `Time`, and `Month`, which represent city areas, weather conditions, time of day, and months, respectively.

The variable `inHours` specifies the time horizon after which the probability of a traffic jam is evaluated.

The static predicate `winter` specifies the winter months.

The instruction `input(curA, curM, curT, curW, inHours)` simulates a request from the controller responsible for the information board to determine whether a congestion warning should be displayed in `inHours` hours for a given area `curA`, current month `curM`, current time `curT`, and weather condition `curW`.

The outer `if` statement verifies the correctness of the input data. For simplicity, we assume that adding `inHours` to the current time does not cause a transition to the next day.

The definition of the dynamic predicate `trafficJam` filters facts from the database. It fixes the specific values `curA` and `curW` for the area parameter `a` and the weather parameter `w`, allows the time parameter `t` to vary within a one-hour interval around the checked moment `curT + inHours`, and restricts the month parameter `m` to winter months only.

The inner `if` statement determines whether a traffic jam is expected in the specified area after `inHours` hours and sends the appropriate message to the controller for display on the information board.

When this program is executed repeatedly within the digital twin, it can simulate the actual probability distribution of messages appearing on information boards in each area, since it relies on real and continuously updated data stored in the database.

## 4.2 EXAMPLE 2: EMERGENCY RESOURCE ALLOCATION (DECISION MODE)

A city's emergency dispatch centre must decide how many ambulances to allocate to each district for the next shift. This decision depends on the probability of incidents occurring in each district, which can be estimated from historical data and real-time signals such as weather conditions, shift, and month. The goal is to minimise response time while maintaining adequate coverage. A decision-support system based on a probabilistic model can estimate these probabilities and recommend resource allocations.

The program below computes the probability of an incident in each district. It uses the dynamic predicate `incident`, which represents statistics of accidents in winter depending on the district, month, and shift. The evaluation mode is set to `decision`, so evaluating a dynamic predicate returns its probability (computed by counting true and false entries in the database over its domain) rather than generating a random sample.

```
#pmode decision;

north := 1; center := 2; south := 3;
District := [north:south];
rainy := 1; normal := 2; snowy := 3;
Weather := [rainy:snowy];
Month := [1:12];
Second := [1:60*60*24]
day = 1;
night = 2;
Shift := [day:night];

sp winter(month) := month = 12 | month = 1 | month = 2;

input(curM, shift, curW);

if(winter(curM) & shift in Shift &
    curW in Weather)
then {
  for d in District do {
    dp incident(d, m, sh, w, s) :
      winter(m) & sh = shift & w = curW & s in Second;

    output("The probability of an accident in district ", d,
           " = ", incident(d, curM, shift, curW, 0));
  }
}
else output("The data is incorrect.")
```

When the program runs in decision mode, evaluating `incident` returns a probability (for example, 0.7 if 7 out of 10 entries are true). This value reflects the empirical likelihood of an incident occurring in the district under the current weather conditions and shift during winter.

Naturally, this example represents a simplified situation. In a more realistic setting, the model would incorporate additional parameters, and the predicate signature and domains would be correspondingly more complex.

The output of the program could, for example, be as follows:

```
"The probability of an accident in district 1 = 0.3"
"The probability of an accident in district 2 = 0.7"
"The probability of an accident in district 3 = 0.2"
```

These probabilities inform the dispatch centre about which districts are most likely to require ambulances. A simple decision rule could allocate resources proportionally to these probabilities. For instance, one could send 58% $(0.7/(0.7 + 0.3 + 0.2) * 100)$ of available ambulances to the center, 25% $(0.3/(0.7 + 0.3 + 0.2) * 100)$ to the north, and 16% $(0.2/(0.7 + 0.3 + 0.2) * 100)$ to the south.

The decision mode provides a direct probabilistic estimate without requiring repeated sampling, making it suitable for real-time decision support.

The database is continuously replenished with observational data in the form of true and false values of the dynamic predicate `incident(d, m, sh, w, s)`.

Database updates can also be modeled using another program written in the $\Delta$P language, which contains instructions for assigning values to dynamic predicates for specific argument combinations.

For example, the program

```
incident(north, 1, day, rainy, 12) := true;
incident(center, 4, night, snowy, 20) := true;
incident(north, 1, day, rainy, 12) := false;
incident(center, 4, night, snowy, 20) := undef;
```

first assigns the value `true` to the predicate `incident` at the points `(north, 1, day, rainy, 12)` and `(center, 4, night, snowy, 20)`. It then changes the value at `(north, 1, day, rainy, 12)` to `false` and removes the value at `(center, 4, night, snowy, 20)` from the database.

## 4.3 SUMMARY

These two examples illustrate how the two evaluation modes of $\Delta$P serve different purposes:

- **Simulation mode** is well suited for generating possible future scenarios and exploring a range of outcomes, which is useful for digital twins and what-if analysis.
- **Decision mode** computes exact probabilities from the database, supporting risk assessment and resource allocation.

Both examples rely on the core constructs of the language: static and dynamic predicates, domain definitions, database updates, and built-in probabilistic semantics. The compact notation and clear separation between evaluation modes make $\Delta$P a convenient tool for prototyping probabilistic models in smart city applications.

## 5 RELATED WORKS

Since the proposed probabilistic language combines both imperative and logical paradigms, we consider related work in three areas: imperative probabilistic languages, logical probabilistic languages, and languages supporting dynamic probabilities.

First, we review work on imperative probabilistic languages. The pWhile language Wiklicky (2016) was, prior to our work, the only imperative language supporting dynamic probabilities. In pWhile,

program execution is controlled using the `choose` statement rather than the probabilistic `if` statement used in our language. The `choose` statement selects one of two code blocks for execution according to dynamically changing probabilities. The formal semantics of the imperative probabilistic language HeyVL Schröer et al. (2023) demonstrates that the case of static probabilities has been studied sufficiently to enable deductive verification techniques. However, the case of dynamic probabilities in imperative languages remains insufficiently explored. Another difference between these approaches and our work is the absence of a logical probabilistic programming component.

Next, we consider work on logical probabilistic languages. The bSystem implements logical rule inference from machine learning models using ontologies Gavrilin & Mantsivoda (2025). The Probabilistic Law Discovery approach Demin & Ponomarev (2023); Vityaev & Smerdov (2007) enables the inference of probabilistic logical rules based on a probabilistic Herbrand basis. Research on probabilistic inductive logic programming Riguzzi et al. (2014) focuses on discovering probabilistic logic programs that maximize probability for positive samples while minimizing probability for negative samples. The QA-RiskPanel system Yakhyaeva et al. (2021) represents an implementation of fuzzy logic that incorporates probabilistic reasoning. Automatic inference in fuzzy logic is supported by the FASILL language implementation Julián-Iranzo et al. (2020). Compared with our approach, these studies on probabilistic logical languages generally do not incorporate the imperative probabilistic programming paradigm.

Since the proposed language is hybrid, we also consider research on hybrid logical systems. One example is a programming language designed to integrate machine learning with symbolic artificial intelligence, in which all program constructs are represented as tensors Domingos (2025). Another approach focuses on improving the reliability of trusted artificial intelligence systems by incorporating a logical component into the platform architecture Nechesov et al. (2025a). A different direction is the oracle-based approach Nechesov et al. (2025b), which separates control logic from program code. This approach has been implemented in the d0sl language Gumirov et al. (2019); Palchunov & Vaganova (2021). In contrast to these approaches, the logical component of our language is designed to support the explicit representation of a knowledge base.

Within the framework of the developed formal language $\Delta P$, the mathematical apparatus of both static and dynamic predicates is applied. The latter are defined analytically based on the static ones using corresponding formulaic expressions. Under the conditions of the Decision mode of program code execution, a strict need arises to calculate probabilistic truth values for dynamic predicates. The algorithmic rules governing the calculation of these metrics directly form the theoretical basis for the logic of probabilistic reasoning within the system.

One of the common theoretical and methodological approaches in this field is the paradigm of fuzzy logic, based on a graded interpretation of the "truth" category. The mathematical tools of this concept rely on a functional principle of calculation: the degree of truth of a complex logical formula is strictly determined by the truth values of its constituent atomic subformulas Hajek (1998).

Such a methodology has demonstrated high efficiency in solving problems related to formalizing the semantic uncertainty inherent in natural language structures. However, when applied to the mathematical modeling of complex heterogeneous infrastructures, such as Smart City systems, this method reveals significant inadequacy. The fundamental vulnerability of this approach is that the initial empirical information undergoes a preliminary transformation—fuzzification (conversion into a fuzzy format)—and only then are the computational procedures initiated. The consequence of this algorithm is the irretrievable loss of a portion of the primary data, which may be of critical importance in the context of the task. As a result, deep logical paradoxes arise, creating serious methodological barriers to the correct and consistent description of subject domains.

As a clear illustration of such contradictions, it can be noted that the truth values for classical tautologies of the form $(\varphi \vee \neg\varphi)$ (the law of excluded middle) and $(\varphi \rightarrow \varphi)$ can be different from $1$. Similarly, the truth of the conjunction $(\varphi \& \neg\varphi)$ (the law of non-contradiction) can differ from $0$. Moreover, some classically equivalent logical formulas often receive differentiated truth evaluations.

For a more profound justification of this thesis, it is advisable to analyze in detail the basic properties of the most common fuzzy logic systems.

*I. Zadeh's max-min logic:*

$$\mu(\neg\varphi) = 1 - \mu(\varphi), \mu(\varphi \& \psi) = \min\{\mu(\varphi), \mu(\psi)\}, \mu(\varphi \vee \psi) = \max\{\mu(\varphi), \mu(\psi)\}.$$

Examples of paradoxes: Let the formula $\varphi$ have the truth value $\mu(\varphi) = 0,5$. Then

$$\mu(\varphi \& \neg\varphi) = 0,5 \ \ and \ \ \mu(\varphi \vee \neg\varphi) = 0,5.$$

*II. Łukasiewicz logic:*

$$\mu(\neg\varphi) = 1 - \mu(\varphi), \mu(\varphi\&\psi) = \max\{0, \mu(\varphi) + \mu(\psi) - 1\}, \mu(\varphi \vee \psi) = \min\{1, \mu(\varphi) + \mu(\psi)\}.$$

Examples of paradoxes: Let the formula $\varphi$ have the truth value $\mu(\varphi) = 0,5$. Then

$$\mu(\varphi\&\varphi) = 0 \ \ and \ \ \mu(\varphi \vee \varphi) = 1.$$

Let the formulas $\varphi, \psi, \xi$ have the truth values $\mu(\varphi) = \mu(\psi) = \mu(\xi) = 0,5$. Then

$$\mu((\varphi\&\psi) \vee \xi) = 0,5 \ \ and \ \ \mu((\varphi \vee \xi)\&(\psi \vee \xi)) = 1.$$

*III. Probabilistic logic:*

$$\mu(\neg\varphi) = 1 - \mu(\varphi), \mu(\varphi\&\psi) = \mu(\varphi)\mu(\psi), \mu(\varphi \vee \psi) = \mu(\varphi) + \mu(\psi) - \mu(\varphi)\mu(\psi).$$

Probabilistic logic possesses the paradoxes of both Zadeh's logic and Łukasiewicz logic.

Within the framework of the formal language $\Delta$P, an implementation of a *semantic approach* to processing imprecise and fragmentary (incomplete) information is proposed, the methodological basis of which is the mathematical apparatus of blurry model theory Yakhyaeva (2025). This theory represents a natural conceptual generalization of classical model theory. In strict contrast to functional fuzzy logics, which are characterized by the violation of basic logical laws, blurry model theory invariantly preserves all fundamental model-theoretic identities. Consequently, it acts as a conservative extension of classical model theory, providing a unique opportunity for parallel and logically consistent operation with both traditional (cricp) and blurry models in a single computational space.

The fundamental difference in the mathematical basis of the proposed methodology is that, at the initial stages of the computational process, the system abstracts away from numerical probabilistic estimates of the occurrence of specific events in the subject domain under study. Instead, the sets of precedents (situations) in which these events were actually recorded are directly used as primary operands.

According to this paradigm, the entire array of available information — including both a formalized description of general patterns in the subject domain (a priori knowledge) and a set of raw empirical data — is first subjected to comprehensive analytical processing in its original form. Only at the final stage is the aggregated result of the computations subjected to the fuzzification procedure, that is, transformed into numerical indicators belonging to the continuous interval $[0, 1]$. Such an algorithmic strategy ensures that at all intermediate stages of logical inference, the system operates with authentic and absolutely relevant data, eliminating the risk of their semantic distortion or loss, which inevitably arise during premature quantification and digitization.

In addition to the works mentioned above, a significant body of research has been devoted to the development of general-purpose probabilistic programming languages. One of the earliest examples is the Church language Goodman et al. (2012), which extends a Lisp-like functional programming paradigm with stochastic primitives and supports Bayesian inference through generative models. Similar ideas are implemented in the Anglican language Wood et al. (2014), which provides a probabilistic programming framework built on top of Clojure and focuses on efficient inference algorithms for complex probabilistic models.

Another influential direction is represented by systems designed primarily for statistical modeling and probabilistic inference. For example, the Stan probabilistic programming language Carpenter et al. (2017) provides a declarative framework for specifying probabilistic models and performing Bayesian inference using advanced sampling methods such as Hamiltonian Monte Carlo. These languages focus primarily on statistical modeling rather than on integrating probabilistic reasoning directly with imperative program control structures.

There are also approaches that combine probabilistic reasoning with logic programming. The ProbLog system De Raedt et al. (2007) extends Prolog by associating probabilities with facts and

rules, enabling probabilistic inference over logical programs. Similar developments include languages and frameworks for statistical relational learning, where logical relations and probabilistic dependencies are modeled simultaneously.

Despite the wide variety of probabilistic programming approaches, most existing systems emphasize either probabilistic modeling in functional or declarative paradigms, or probabilistic extensions of logic programming. In contrast, the proposed $\Delta$P language aims to integrate probabilistic reasoning with imperative program control and a logical knowledge representation component within a unified framework. This combination enables the description of complex computational processes that involve both algorithmic control flow and probabilistic logical reasoning.

Thus, the main advantage of our language lies in the synergy of the following features: combining imperative, logical, and probabilistic aspects within a single language; providing the ability to vary the methods used to compute the probabilistic semantics of logical operations; dynamically estimating predicate probabilities through a database with facts filtered according to the predicate's domain; and offering a choice of evaluation mode – either oriented toward multiple program executions (as used in digital twins) or toward a single execution that produces probabilistic recommendations (as used in decision-support systems).

## 6 Conclusion

This paper presented the $\Delta$P programming language, a probabilistic extension of an imperative programming framework designed to support reasoning under uncertainty. The language integrates deterministic programming constructs with probabilistic predicates, enabling developers to express both logical conditions and stochastic relationships directly within programs.

Currently, $\Delta$P is best understood as a conceptual framework for combining imperative, logical, and probabilistic aspects within a single language, with several distinctive features. It provides the ability to vary the methods used to compute the probabilistic semantics of logical operations and to dynamically estimate predicate probabilities through a database, with facts filtered by the predicate's domain and with a choice of evaluation mode: either oriented toward multiple executions of a program (as used in digital twins) or toward a single execution that produces probabilistic recommendations (as used in decision-support systems).

This framework is presented through a description of a minimal set of language constructs, a specification of a formal small-step operational semantics for these constructs, and is illustrated with two small examples, one for each evaluation mode.

In future work, the plan is to expand the set of constructs into a full-fledged language, including a library, implement the language, and test it on real-world examples.

Future work may also include developing static analysis and verification techniques for probabilistic programs, and implementing efficient runtime systems and tools for program development and debugging. Further research may also explore integration with machine learning frameworks and other probabilistic modeling approaches.

### Acknowledgments

Sergey Goncharov and Evgenii Vityaev was financially supported by the State Assignment of the Sobolev Institute of mathematics SB RAS for 2026-2029 "Theory of computability and logical aspects, logical calculus and their semantics". Supervisor: S.S. Goncharov. Project Number: FWNF-2026-0032.

This work was supported by a grant for research centers, provided by the Ministry of Economic Development of the Russian Federation in accordance with the subsidy agreement with the Novosibirsk State University dated April 17, 2025 No. 139-15-2025-006: IGK 000000Ts313925P3S0002.

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
