# OpenReview forum: "THE ∆P PROGRAMMING LANGUAGE: COMBINING IMPERATIVE, LOGICAL, AND PROBABILISTIC CAPABILITIES"
_mathai.club/MathAI/2026/Conference — 2026 Oral_

### Official Review · Reviewer_115z · 2026-03-11
**Review of "THE ∆P PROGRAMMING LANGUAGE: COMBINING IMPERATIVE, LOGICAL, AND PROBABILISTIC CAPABILITIES"**

**Rating:** 6
**Confidence:** 2

**Review:**

The paper introduces the **∆P programming language**, which aims to integrate *imperative programming*, *logical reasoning*, and *probabilistic inference* within a single computational framework. The motivation of the work is to support reasoning under uncertainty directly at the programming language level. The authors describe the syntax and structure of ∆P programs, including **terms**, **predicates**, **formulas**, **declarations**, and **control constructs**, and provide a conceptual description of the operational semantics. In the proposed framework programs may contain both deterministic and probabilistic assertions, and program execution can be interpreted as a reasoning process over logical formulas associated with probabilities. A probabilistic predicate can be formally interpreted as a mapping $( P : \mathcal{F} \rightarrow [0,1] \), where \( \mathcal{F} \) $denotes the set of logical formulas in the program and $\( P(\varphi) \)$ represents the probability that the formula $\( \varphi \)$ holds. The paper presents several small examples illustrating how the language could be used to express uncertain knowledge and reasoning tasks. The topic of the paper is relevant since combining programming languages with probabilistic reasoning is an important direction in modern AI and knowledge-based systems, especially in settings where uncertainty and incomplete information must be modeled explicitly. The paper also provides a reasonably structured presentation of the language design and the motivation is generally clear. The attempt to combine deterministic computation with probabilistic reasoning within a unified framework is conceptually interesting and aligns with ongoing research in *probabilistic programming* and *probabilistic logic systems*. However, the work has several limitations. First, the level of novelty appears limited because similar ideas have been explored extensively in probabilistic programming languages and probabilistic logic programming frameworks. The paper does not clearly explain what fundamentally distinguishes **∆P** from existing approaches. Second, the discussion of related work is limited and does not provide a systematic comparison with existing systems, making it difficult to evaluate the true contribution of the proposed language. Third, the probabilistic semantics are not described in sufficient formal detail. In particular, it is unclear how probabilities propagate through logical operators such as conjunction or implication. For example, in probabilistic reasoning one usually needs formal definitions for expressions such as $\( P(A \land B) \)$ or conditional probabilities defined as $\( P(A \mid B) = \frac{P(A \land B)}{P(B)} \)$, but the paper does not specify how such quantities are computed within the proposed language or whether independence assumptions are used. Another limitation is the absence of an implementation or experimental evaluation. The paper does not describe a **prototype interpreter**, **compiler**, or any empirical experiments demonstrating the practical benefits or performance of the language. In addition, the examples included in the paper are relatively simple and do not clearly demonstrate advantages of the proposed approach for complex reasoning tasks or real-world applications. In terms of clarity, the paper is generally readable and the main ideas can be followed, but several definitions remain informal and the probabilistic semantics could be presented in a more rigorous way. Providing more formal definitions, theoretical properties, or implementation details would strengthen the contribution. Overall, the paper addresses an interesting topic and proposes a conceptual framework for integrating probabilistic reasoning into a programming language, but the current version lacks sufficient novelty, detailed formalization, comparison with prior work, and empirical validation.

---

> ### Author Rebuttal · Authors · 2026-03-13
>
> Part 1
> - Dear Reviewer,
> - On behalf of all authors, I would like to thank you for your thoughtful comments and valuable suggestions regarding our manuscript. We have found your feedback extremely helpful in improving the quality of our work.
> - Below is a summary of our changes according to your recommendations:
> 	- **The paper does not describe a prototype interpreter, compiler, or any empirical experiments demonstrating the practical benefits or performance of the language.**
> 		- We fully acknowledge the importance of this suggestion. This work will undoubtedly be carried out in the future. Unfortunately, we did not clearly and explicitly state the idea that currently, ΔP is best understood as a conceptual framework for combining imperative, logical, and probabilistic aspects within a single language, with several distinctive features. The corresponding clarifications have been added to the Сonclusion Section  (page 14, lines 726-737).
> 			- Currently, ΔP is best understood as a conceptual framework for combining imperative, logical, and probabilistic aspects within a single language, with several distinctive features. It provides the ability to vary the methods used to compute the probabilistic semantics of logical operations and to dynamically estimate predicate probabilities through a database, with facts filtered by the predicate’s domain and with a choice of evaluation mode: either oriented toward multiple executions of a program (as used in digital twins) or toward a single execution that produces probabilistic recommendations (as used in decision-support systems).
> 			- This framework is presented through a description of a minimal set of language constructs, a specification of a formal small-step operational semantics for these constructs, and is illustrated with two small examples, one for each evaluation mode.
> 			- In future work, the plan is to expand the set of constructs into a full-fledged language, including a library, implement the language, and test it on real-world examples.

---

> > ### Author Rebuttal · Authors · 2026-03-13
> >
> > Part 2 (Continuation)
> > - Below is a summary of our changes according to your recommendations:
> > 	- **In addition, the examples included in the paper are relatively simple and do not clearly demonstrate advantages of the proposed approach for complex reasoning tasks or real-world applications.**
> > 		- We fully agree with you regarding real-world examples, and they will be addressed in the future. However, ΔP is only a conceptual framework, so these toy examples simply illustrate the main constructs of the language (static and dynamic predicate definitions, modes, and so on) and their expressive power.
> > 	- **In terms of clarity, the paper is generally readable and the main ideas can be followed, but several definitions remain informal and the probabilistic semantics could be presented in a more rigorous way.**
> > 		- The probabilistic semantics of logical operations can be configured via macros. However, if macros are not defined in the program, the default probabilistic semantics are used. The default values are described in Section 2.7 (pages 3-4, lines 161-168). For example, for implication (page 4, lines 167-168), the default is specified as $1 - x + x * y$ , where $x$ and $y$ are the probabilities of the implication’s arguments. The formal operational semantics of logical operations for arbitrary semantics defined via macros are given in Section 3.4 (page 6, lines 289-293).
> > - We hope that the revised version is now more balanced.
> > - Thank you for your time and consideration.
> > - Sincerely,
> >   Anonymous Author

---

### Official Review · Reviewer_emJJ · 2026-03-12
**This paper proposes ∆P, a programming language that integrates imperative programming with probabilistic reasoning and logical predicates. The authors define its syntax and semantics and illustrate its use through example decision-making scenarios under uncertainty.**

**Rating:** 5
**Confidence:** 3

**Review:**

This paper proposes ∆P, a probabilistic extension of an imperative programming language that aims to combine imperative control flow, logical predicates, and probabilistic predicates in one framework. The paper presents the language syntax, macro system, and a small-step operational semantics, then illustrates usage through two smart-city style case studies: traffic flow prediction in simulation mode and emergency resource allocation in decision mode. The intended contribution is a unified framework for reasoning under uncertainty inside imperative programs.

Strengths:

The paper has a clear high-level motivation and tackles a meaningful problem: many systems need both algorithmic control and explicit treatment of uncertainty, and the paper attempts to bridge that gap in a single formalism. A positive aspect is that the authors do not stop at an informal language sketch; they provide explicit language constructs and operational semantics, which gives the work a more rigorous foundation than a purely conceptual proposal. The two execution modes, decision and simulation, are also easy to understand and help communicate the intended use cases.

Weaknesses:

The main limitation is that the paper feels more like an early language proposal than a mature research contribution. Although syntax and semantics are defined, there is no implementation, no interpreter or compiler, no experimental evaluation, and no comparison on concrete tasks against existing probabilistic programming or logic-based systems. The conclusion itself frames efficient runtime systems, debugging tools, static analysis, and ML integration as future work, which suggests that the practical contribution is still incomplete.

A second concern is that the case studies are not yet polished enough to convincingly validate the language. The examples are illustrative, but they remain toy demonstrations rather than evidence of expressiveness, efficiency, or usability. More importantly, the manuscript contains visible drafting and consistency issues inside the examples, including a leftover “todo” note, inconsistent or questionable variable usage, and code-like fragments that appear syntactically problematic. For instance, the traffic example still contains todo: adds in, for, input, channels, uses hours where inHours seems intended, and has an output line with mismatched punctuation; the ambulance example defines for loc in District but outputs probabilities using curD, which weakens confidence in the example’s correctness.

A third issue is novelty positioning. The related work section cites several relevant paradigms and argues that ∆P combines imperative, logical, and probabilistic components, but the differentiation remains mostly descriptive. The paper would be stronger if it made a sharper technical case for what ∆P can express or verify that competing approaches cannot, or if it provided a formal expressiveness/result beyond the semantics definitions themselves. As written, the novelty is plausible but not yet convincingly demonstrated.


Recommendations:

The paper would improve substantially with three additions. First, the authors should provide a minimal implementation or prototype interpreter and evaluate it on at least a few nontrivial tasks. Second, the examples should be carefully cleaned and verified so that they function as trustworthy demonstrations of the language rather than rough sketches. Third, the paper needs a stronger empirical or formal comparison with existing probabilistic and logic-based languages, ideally showing what ∆P uniquely enables in practice or in theory.

---

> ### Author Rebuttal · Authors · 2026-03-13
>
> Part 1
> - Dear Reviewer,
> - On behalf of all authors, I would like to thank you for your thoughtful comments and valuable suggestions regarding our manuscript. We have found your feedback extremely helpful in improving the quality of our work.
> - Below is a summary of our changes according to your recommendations:
> 	- **First, the authors should provide a minimal implementation or prototype interpreter and evaluate it on at least a few nontrivial tasks**
> 		- We fully acknowledge the importance of this suggestion. This work will undoubtedly be carried out in the future. Unfortunately, we did not clearly and explicitly state the idea that currently, ΔP is best understood as a conceptual framework for combining imperative, logical, and probabilistic aspects within a single language, with several distinctive features. The corresponding clarifications have been added to the Сonclusion Section  (page 14, lines 726-737).
> 			- Currently, ΔP is best understood as a conceptual framework for combining imperative, logical, and probabilistic aspects within a single language, with several distinctive features. It provides the ability to vary the methods used to compute the probabilistic semantics of logical operations and to dynamically estimate predicate probabilities through a database, with facts filtered by the predicate’s domain and with a choice of evaluation mode: either oriented toward multiple executions of a program (as used in digital twins) or toward a single execution that produces probabilistic recommendations (as used in decision-support systems).
> 			- This framework is presented through a description of a minimal set of language constructs, a specification of a formal small-step operational semantics for these constructs, and is illustrated with two small examples, one for each evaluation mode.
> 			- In future work, the plan is to expand the set of constructs into a full-fledged language, including a library, implement the language, and test it on real-world examples.
> ...

---

> > ### Author Rebuttal · Authors · 2026-03-13
> >
> > Part 2 (Continuation)
> > - Below is a summary of our changes according to your recommendations:
> > 	- **Second, the examples should be carefully cleaned and verified so that they function as trustworthy demonstrations of the language rather than rough sketches.**
> > 		- The artifact “todo: adds in, for, input, channels,” which served as a reminder of constructs that should be added to the language and have their formal semantics specified, has been removed. All of these constructs -- except for channels, which are left for a future extension of the language -- have now been introduced into the language, and their formal semantics have been defined.
> > 		- The variable hours has been replaced with inHours in the derivation.
> > 		- The confusion involving loc and CurD has been corrected.
> > 	- **Third, the paper needs a stronger empirical or formal comparison with existing probabilistic and logic-based languages, ideally showing what ∆P uniquely enables in practice or in theory.**
> > 		- We added a summary in the Related Works section (page 14, lines 711-717) that more precisely positions our work relative to the studies discussed in that section.
> > 			- Thus, the main advantage of our language lies in the synergy of the following features: combining imperative, logical, and probabilistic aspects within a single language; providing the ability to vary the methods used to compute the probabilistic semantics of logical operations; dynamically estimating predicate probabilities through a database with facts filtered according to the predicate’s domain; and offering a choice of evaluation mode -- either oriented toward multiple program executions (as used in digital twins) or toward a single execution that produces probabilistic recommendations (as used in decision-support systems).
> > - We hope that the revised version is now more balanced.
> > - Thank you for your time and consideration.
> > - Sincerely,
> >   Anonymous Author

---

### Decision · Program_Chairs · 2026-03-14

**Decision:**

Accept (Oral)

**Comment:**

Dear Author(s),

On behalf of the Program Committee of the International Conference on Mathematics of Artificial Intelligence (MathAI 2026), we are pleased to inform you that your paper has been accepted for an oral presentation at MathAI 2026.

Your paper was evaluated through a rigorous two-stage review process involving both automated screening and expert review by members of the Program Committee. The reviewers recognized the quality and contribution of your work.

Presentation details:

- Format: Oral presentation (15–20 minutes + 5 minutes Q&A)
- Mode: You may present either in person (offline) at the conference venue in Sirius, Russia, or remotely via Zoom. Please indicate your preferred mode when confirming your participation.
- Conference dates: Marh 30 - April 3, 2026
- Website: https://mathai.club

Next steps:

1. Please confirm your participation and presentation mode by replying to this email mathai.club@yandex.ru no later than March 15, 2026 18:00 Moscow time.
2. If you plan to attend in person, the organizing committee will provide accommodation details separately.
3. Please prepare your final camera-ready manuscript according to the formatting guidelines available at https://mathai.club and upload it to OpenReview by March 15, 2026 18:00 Moscow time.

Should you have any questions regarding the program, logistics, or your presentation slot, please do not hesitate to contact us.

We look forward to your contribution to MathAI 2026.

With kind regards,

MathAI 2026 Program Committee
International Conference on Mathematics of Artificial Intelligence
https://mathai.club
OpenReview: https://openreview.net/group?id=mathai.club/MathAI/2026/Conference
Telegram: https://t.me/MathAI_club
Email: mathai.club@yandex.ru